# Sustainable Strategic People Management: A Confucian Perspective on Chinese Management

Yingying Zhang-Zhang

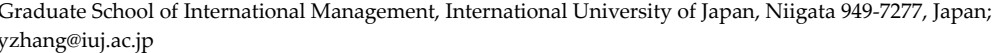

Graduate School of International Management, International University of Japan, Niigata 949-7277, Japan; yzhang@iuj.ac.jp

**Abstract:** This paper examines the strategic management of people within enterprises as a driver of sustainable growth. As strategic people management (SPM) is founded on the Eastern knowledge workers' perspective, we integrate SPM with the Confucian perspective to analyze the factors underlying the sustainable success of Chinese management. In so doing, we review the literature on sustainability, SPM, Chinese management, and the integration of Confucian cultural values. We utilize the qualitative case research method to examine 20 successful Chinese enterprises across five sectors. The results of the case analysis reveal three types of sustainable SPM associated with Confucian values: responsible people management, paradoxical people management, and humanistic people management. We propose a conception of sustainable SPM combined with Confucianism to be relevant in the Chinese business context, where a high degree of dynamism can be seen. The findings of this study could be extended through analyses conducted in other contexts with a high level of complexity, such as emerging markets, disruptive technology, unexpected crises, or any aggregated interactions of such contexts.

**Keywords:** SPM; management by values; VUCA; dynamic environment; SHRM; Confucianism; sustainability; China

---

## 1. Introduction

The increasing prominence of the Chinese economy in global business has prompted much interest and curiosity among policymakers, business practitioners, and scholars seeking to better understand the phenomenon of Chinese management [1]. The focus is not only on China's sustainable gross domestic product (GDP) growth over the last three decades, which has led the country to assume a top economic power position in terms of GDP based on the purchasing power parity (PPP) value [2], but also on how China has become a reference point for emerging markets seeking to benchmark their sustainable growth in terms of the economic dimension. Different from many other emerging economies considered sources of cheap production, over the last decade China has transformed itself from the world's manufacturing center to an innovator, converting itself into the global leader with regard to intellectual property (IP) applications due to its continuous sustainable growth [3–5].

While there remain social, political, and economic issues that China needs to resolve to ensure its sustainable development, the country's sustainable growth over the last few decades is evidenced by factual data and largely attributed to both the development of the private business sector and institutional factors [6,7]. When seeking to understand Chinese management, it is important to recognize that culture and leadership are among its distinctive features [7–9].

Management in China, as a nation rising from the East, shares certain similarities with Japanese management due to Japan having developed its economy and emerged as a global power after the Second World War. Both countries also share Confucian roots in values and strongly orient their management beliefs based on people-centricity. Culture,

as an embedded element influencing how people interpret, behave, and act, has been especially relevant in terms of people management due to the fact that people may perceive and behave differently toward the same organizational strategy and policies of an organization or enterprise in a global context. Hofstede and Bond [10] establish the correlation between Confucianism and the economic boom in the Southeast Asia region, and we pursue a similar line of thought to examine the concept of sustainable strategic people management (SPM) [11] based on the cultural values of Confucianism. The Chinese management philosophy and the associated people management practices within firms represent the engine driving the economic dimension of China's sustainable growth [7]. The innovation transformation that has been ongoing over the last few decades and has driven the upswing of the Chinese economy and its sustainability has also demonstrated that Confucian culture underlies and nurtures the development of creativity and innovation [5].

The remainder of this paper is organized as follows. First, the theoretical background section presents a critical review of the SPM theory and its relevance to sustainability in a highly dynamic context, followed by an examination of the evolution of Chinese firms, the concept of people management, and the relations between SPM and Confucianism. The methodology section then describes the case selection, data collection, and analysis. After the results are presented, they are discussed and their implications addressed, including consideration of future research directions concerning sustainable management, which may be especially useful for emerging market contexts.

## 2. Theoretical Background

### 2.1. Sustainability and Strategic People Management

It remains debatable whether sustainability is a fad or an example of rhetoric in the field of management, despite its increasing impacts and the growing interest in it on the part of scholars and practitioners [12]. No matter which argument is preferred, the fact is that sustainability is a valuable currency in a field with global concerns. Despite them being considered "grand challenges", many organizations and institutions are striving to integrate the United Nations' sustainable development goals (SDGs) into strategies, policies, and practices for implementation in many dimensions [13]. As a global concern, multinational corporations (MNCs), whether large or small and medium-sized enterprises, are also making efforts to play a role and act in relation to sustainability and the SDGs, regardless of critiques concerning their insufficiency [14–19].

While sustainability has become a fashionable topic in the fields of management and international business, it can be divided into three dimensions, namely the economic, social, and environmental or ecological dimensions [20], with enterprises from both advanced economies and emerging markets being affected by those dimensions [21]. The economic dimension of sustainability has always been a concern of firms in terms of their management practices, as firms inherently need to sustain their economic results to ensure their financial performance and longevity, although the social and ecological dimensions are relatively recent subjects of managerial attention, especially when doing business in developing countries and emerging markets [15,22,23]. The factors that underlie the effective implementation of sustainability strategies in emerging markets largely rely on the commitment of top management and discretionary slack, in addition to external factors such as legal forces [23]. Therefore, the value represented by leaders and the ability to spur on managers and staff at different levels are essential in terms of allowing effective strategic management to function within enterprises [7], including their sustainability strategies.

The conceptualization of SPM proposes sustainability as a strategic orientation rather than a performance measure, as it is perceived within the traditional focus. Both leadership and cultural values are incorporated as key constituents of the achievement of the multiple dimensions of sustainability in a turbulent and highly dynamic context [11]. In terms of a highly dynamic context, a specific strategy may require constant adjustments in accordance with environmental, knowledge-related, and innovation management changes if it is to become more critical in such a context. While dynamic capabilities must be able to

flow (i.e., to sense, organize, capture, and renew knowledge and information, to generate new knowledge and organization), relatively static organizational elements such as structure and specific human resources (HR) practices may be unable to cope due to the high costs associated with constantly changing them, whereas leadership, culture, and learning could be the dynamic elements required to facilitate the contributions of people to the formulation of strategies [24].

Emerging markets are often characterized by high levels of volatility, uncertainty, complexity, and ambiguity (VUCA), which are considered sources of highly dynamic contexts, along with other elements such as technology disruption and unexpected crisis events [11,25,26]. China, as a leading emerging market, is characterized by all these factors and their interactive effects. In addition to learning technology from the West following its economic opening, China has also developed its own technologies and innovations, which have positioned it as the number one country in terms of IP applications since 2012. Certain other emerging economies, such as India, Turkey, Iran, and Russia, are also rising to the top of the global ranking with regard to IP [3–5]. Chinese enterprises such as Alibaba, Tencent, and PingAn are among the top-ranked patent grantees in relation to the emerging blockchain technology associated with financial technology (fintech) and artificial intelligence (AI) technology [27]. Moreover, significant crises such as the 1998 Asian financial crisis and the 2008 global financial crisis have served as opportunities for China to become more resilient. In light of all this, how do Chinese enterprises manage to enhance their innovation capability over time and ensure their development, thereby sustaining the nation's growth?

### 2.2. Chinese Firms and People Management

From the outset, it is important to understand the economic transformation that China and Chinese management have been experiencing, particularly its implications for research concerning people management and SPM development. According to the International Monetary Fund (IMF), in terms of GDP based on the PPP value, China overtook the United States and became the world's largest economy in 2016, and it has maintained its number one position since then [2]. This development was a result of the Chinese economic miracle, which stemmed from the country's capability to consistently grow at a rate of approximately 10% per year. Despite the growing base for continuous development and the fact that it is difficult to maintain a two-digit growth rate, the Chinese GDP growth rate has been maintained at a relatively high ratio. This has even proven true during the COVID-19 pandemic because, in spite of a predicted economic slowdown, the factual and projected data concerning China are still much more optimistic than the data concerning advanced economies provided in the World Economic Outlook overview [28].

In an attempt to predict China's economic growth, some speculate that the post-COVID-19 rebound will result in the effects of the trade war, decoupling, and deglobalizing [29,30]. In earlier studies, some scholars argued that the Chinese economic rise is a result of different paths: top-down, centrally managed capitalism and bottom-up, decentralized grass-roots capitalism [6,31]. During the process of transitioning from a planned economy to a market economy, Chinese privately-owned enterprises (POEs) have benefited from the related economic opening policies, along with other ownership modes such as state-owned enterprises (SOEs) and foreign-owned enterprises (FOEs), although the former has been less protected and supported than other types of enterprises [7]. This is due to the so-called "visible hands" of the leadership, which manage the success of POEs in China's highly competitive markets as well as in global markets. In the Chinese domestic markets, SOEs have enjoyed government support with regard to resource access and acquisition, while FOEs have the privilege of negotiating with government agencies at different levels as well as fiscal advantages. For their part, POEs can only rely on their managerial capability, social networking, and entrepreneurial spirit to address all the challenges (e.g., less chance to access bank loans) found on the less trodden road at the beginning of entrepreneurial exploration in China [7].

Due to the rise of the Chinese economy, both Chinese enterprises and Chinese management have been the subject of considerable attention, which has led to the integration of diverse Western and Japanese managerial models with traditional Chinese practices, especially with regard to the dominant management theories that originated in the United States [32]. Aside from the dominance of Western theory in the Chinese context, there have also been indigenous attempts to unfold the paradoxical Chinese model in order to complement or move beyond established frameworks [33,34]. This hybrid knowledge exchange and flow between the West and the East can be similarly seen in Japanese management studies [35]. Within the Chinese system for managing people, hybrid mechanisms also reflect the evolution of the dynamic tradeoff between macroeconomic, institutional, and organizational factors, a process characterized by mutual learning between the East and the West [36]. This complex and compressed process of industrialization and management that has unfolded in China following its economic opening [37], as well as the reality of the country being on its way to leading with regard to certain digital innovations [25], highlight the coexistence of the old and the new, in addition to the featured potential of China. While institutional factors play an important role [38], people, knowledge, culture, and the intertwined variations in such factors feature the required characteristics of value, rareness, inimitability, and organizability (VRIO) as resources required to sustain firms' competitive advantages [39].

People management has often been identified as important in terms of managing Chinese enterprises due to humanism being its core value [40]. At the macro level, economic reforms have driven the mobility of labor (i.e., people immigrating from rural to urban areas), the termination of the historical "iron rice bowl" (i.e., lifetime employment), and the weakening of the "HuKou" system (i.e., Chinese household registration system) [41]. Gradually, the well-educated workforce has become the source of a talent war being waged among both Chinese enterprises and multinationals operating in China, while cost efficiency is being increasingly considered in relation to work associated with less value-adding tasks. At the micro level, differentiated ways of managing people within enterprises have been developed following the introduction of HR practices originating in the West, such as recruitment, performance appraisal, and the high involvement/high performance work system (HIWS/HPWS), to strategically manage successful businesses in China [9,42]. These old and new ways of managing people, in addition to the coexistence of hybrid employee profiles (e.g., old and young generations, rural and urban populations, international and local talent), have combined to increase the complexity of people management in China [5,43], where dual systems and mechanisms are often applied [44–46]. Yet, while human resource management (HRM) policies and practices from the West have been adopted by Chinese enterprises and adjusted to the Chinese context in an effort to accommodate its uniqueness and its traditional personnel management practices, the effectiveness of such policies and practices has been much questioned in the literature [1,9,31,42,47].

### 2.3. Sustainable Strategic People Management and Confucianism

SPM considers culture to be a key component of its dynamic capability flow. Here, the study of cultural values is quite complex. The idea of cultural studies within organizations was introduced in the 1960s, has evolved over time, and has now been the focus of attention in the management and international business fields for decades [48–56]. One remarkable example in this regard is the work by Hofstede and his various colleagues [56–58]. National culture and organizational culture are often highlighted as the two most relevant levels of study in the international context, and they have certainly driven the evolution of cultural value studies in such a context, which have now entered a stage of complexity [9,55,58]. Management scholars such as Drucker and Chakraborty also highlight the issues of ethics and values in management [59,60]. To make management sustainable, the management by values (MBV) concept suggests three dimensions of values to interact and balance within the organizational space, which represents a different approach from earlier mechanical views of management style, such as management by instructions and management by ob-

jectives [61–63]. MBV considers the economic, ethical, and emotional dimensions of values to represent management philosophy and practices, which could be major drivers of the development of sustainable, competitive, and more humane cultural values within organizations [62]. The MBV conceptualization appears to function well in organizations facing highly dynamic contexts associated with unstable, turbulent, and uncertain environments [64], and consequently, it may grasp and manage the flexible elements that need to be instilled in the minds of people serving as knowledge workers to guide their practices and behaviors in order to achieve the multiple goals of sustainability. Due to being strongly rooted in traditional Chinese values such as Confucianism, Chinese management may sustain its distinctive cultural features through self-renewing processes involving innovative mechanisms [1,5,8,65].

People management is not a new concept, although SPM is a novel conceptualization developed as an extension of strategic human resource management (SHRM) [11]. The agreed and consolidated theoretical framework is still challenging, and the paradigm appears unable to respond to the current strategic needs of the firms despite its flourishing and legitimization in the research field [66–68]. Rather than an extension, it is probably more appropriate to consider SPM a different paradigm from SHRM. As a major point of difference, SPM can be distinguished from SHRM due to having sustainability at the core of its strategic orientation, which it achieves by situating people as the center of attention, while SHRM is performance-oriented and relies on HRM to be profitable [11,69]. Among other aspects, SHRM focuses on aligning HR with strategy implementation [70,71], which is appropriate in a relatively stable environment but difficult to achieve in a highly dynamic environment, such as emerging markets, where VUCA occurs in daily life [22]. While it is true that the strategic importance of people ( 人 in Chinese) has been widely considered and studied in China, meaning that it has its strategic roots in ancient Chinese philosophy, the term HRM may be of concern because the term HR ( 人力资源 in Chinese) did not exist in ancient China. In this sense, people management and HRM need to be explicitly distinguished, as some scholars have sought to do in the contemporary management field [11,72,73].

Rather than narrowly focusing on employees within an organization, ensuring financial performance, and aligning with the firm strategy as in the case of HRM, SPM places people at the center of the firm to ensure its long-term survival and growth with a sustainability orientation. By offering an amplified meaning and a broader boundary, SPM recognizes all ranges of people, that is, stakeholders both inside and outside of the organization, to interact, create, and transfer knowledge (i.e., treating people as knowledge workers who are capable of utilizing their imagination and innovative capability not only to absorb and learn but also to create) [11]. As SPM could significantly contribute to strategy formulation given the importance of knowledge, it works better in highly dynamic VUCA contexts in which uncertainty is certain, which allows it to be more sustainable. Although Sun Tzu is quoted by Losovski concerning the strategic relevance of people in relation to actions [74,75], most of the associated strategic considerations are related to military activities and contexts, which *The Art of War* addresses. In the sphere of management and governance, Confucianism is the most common line of philosophical thought due to its ethical concerns regarding inner virtue, morality, and values [40,65,76,77]. Moreover, Hofstede and Bond also note that Confucianism could be a driver of the economic growth seen in the East Asia region, as all the emerging economies there have similar cultural roots and reflect the influence of Confucianism [10], although some may disagree with this [65]. If East Asian economic rise and success could be attributed to Confucianism, as has been the case for Japan, Korea, Singapore, Hong Kong, and Taiwan, it is arguable that China's rapid economic development could also be highly related to the Confucian culture. In fact, it is arguable that the country's Confucian roots have contributed to China's dynamic cultural shift toward enhanced innovation capability through self-cultivation, lifelong learning, tolerance of mistakes, and moderation [5].

Four books have been compiled by Zhu Xi to represent the philosophy of Confucianism: *The Analects of Confucius*, *The Book of Mencius*, *The Great Learning*, and *The Doctrine of the Mean*. As a philosophy, Confucianism self-evolves and self-renews, with Neo-Confucianism forming the core of Confucian concern for society and government, as synthesized with the two other principal Chinese ancient philosophies, namely Taoist cosmology, and Buddhist spirituality, which predominated in intellectual and spiritual life in China, Korea, and Japan prior to the modern period [78]. Taking human nature as the starting point for the orthodox rules, the Confucian conception of governance not only consists of virtue, benevolence, and righteousness but also of the harmony of humans and nature ( 天人合一 in Chinese) [79]. If the practical significance of Confucian governance is performance-oriented, its people orientation concerns socio-ethical values [79], while the harmony between humans and nature can be viewed as ecological values.

Some aspects of Chinese business culture that have been influenced by Confucianism can be extracted here: "harmony", "conscientiousness and consideration of others", "humanistic management", "ethical issues", "democratic leadership", "investigation (learning)", "knowledge sharing via relations and trust", "reciprocity", and "the use of ritual and time to build trust" [65,80–82]. Nonetheless, above all, it is a form of people-centric management, meaning that people are placed in a strategic position with regard to governance and management. Moreover, Confucianism places ethics and values at the core of all actions: "from which to think, to envision the meaning of our conduct and find guidance for action, policies, and decisions... help us at a point where we sorely need help" [78], in addition to utilizing human emotions such as "jiaoqing" or "ganqing" to persuade each other to engage in knowledge sharing [65]. Furthermore, the two Neo-Confucianism concepts of "ch'i" and "li" represent an energetic thrust toward systematic complexity, emerging and evolving with the complex pattern of organisms and ecosystems in a self-organizing way, starting "at a relatively simple level and transform as adaptive strategies within the system lead to continually increasing levels of complexity" [78] or self-renewing and contributing to innovation [5]. The dynamic nature of SPM requires a dynamic people management capability to function in a highly dynamic and complex environment [11], which echoes the complexity perspective [83] and the dynamic adaptive logic [24] and eventually contributes to sustainable SPM in a highly dynamic context.

## 3. Research Methodology

### 3.1. Case Selection

We utilized 20 successful Chinese enterprises previously collected as cases for our data analysis. The selection criteria are as follows: (1) successful Chinese enterprises, which allows us to observe and interpret their sustainable SPM in multiple dimensions; (2) firms of a significant size and scale instead of small and medium-sized firms, which allows us to observe and interpret the governance impacts; and (3) firms located in different industries, which allows us to observe the sustainable SPM functions across sectors instead of as features of a particular sector.

To reflect the sustainability focus of this study, the measure of success is multi-dimensional. Indeed, rather than solely considering the firms' economic performance, we also consider social dimensions such as social impact and reputation, as well as ecological elements not only concerned with environmental matters but also related to the Confucian sense of the harmony of humans and nature. The selected cases are drawn from five sectors: telecoms and tech, consumer goods and retail, finance and services, digital business, and construction and real estate (see Table 1).

**Table 1.** Distribution of the studied cases.

|  | Selected Cases | Main Business |
|---|---|---|
| Telecoms and Tech (5) | Lenovo<br>Huawei<br>Datang<br>ENN<br>Huayi | Personal computers<br>Telecoms equipment<br>Telecoms technology<br>Clean energy<br>Compressors |
| Consumer Goods and Retail (5) | Li-Ning<br>Wahaha<br>Kidswant<br>Haier<br>Hisense | Sportswear<br>Beverages<br>Kids' goods<br>Electronics goods<br>Electronics goods |
| Finance and Services (4) | Fosun<br>Taikang<br>China Merchant Bank<br>Haidilao | Investment<br>Insurance<br>Banking<br>Restaurant chain |
| Digital Business (3) | Alibaba<br>HC360.com<br>Neusoft | E-commerce<br>E-commerce<br>Software |
| Construction and Real Estate (3) | Vantone<br>Vanke<br>Tecsun | Real estate<br>Real estate<br>Construction |

The 20 selected enterprises are popular in their corresponding sectors in China due to their high performance, leading position, and sustainable growth over decades. Most are highly recognized due to their brand awareness and social achievements in terms of positive impacts on society through ethical behavior or harmonious management. With regard to the enterprises' achievements in different dimensions, most have been collected into a series of books on Chinese management by internationally reputed publisher Palgrave MacMillan [5,7]. Moreover, Alibaba, Huawei, Haier, and Lenovo are among the enterprises that regularly appear in Western media and business news; Fosun, Hisense, Huayi, and Haidilao are active in international business expansion but less popular; while Vantone, Kidswant, and Tecsun mainly focus on Chinese markets [5,7].

### 3.2. Data Collection and Analysis

The case method is especially advantageous for exploratory research. When the phenomenon under study is relatively recent and the theoretical framework is less mature, the exploratory type of qualitative case offers the insights required to derive valid observations and interpretations [84]. The selected cases have been studied over time by the research team to facilitate the data collection. More specifically, the research team collected the case data via secondary data sources and first-hand interviews as part of several projects on Chinese management. Case reports were written and published in different international volumes or in the international case center. This study utilizes reported case data collected from these publicly published materials, that is, two books from the Chinese Management series published by Palgrave MacMillan and one case published in the Ivey Case Center. In addition, whenever necessary, raw data are consulted, and updated data from secondary public sources, such as new publications by reliable sources, company websites, and annual reports, are checked to verify the data [1,5,85].

For the data analysis, the SPM framework and Confucian values are used to codify the data derived from the cases, with the meaning being interpreted at both the manifested and latent levels. The codes are compared across cases, then across sectors, and then clustered into themes and corresponding categories [86,87]. Following the suggestion of Boyatzis [86], the cases are organized into subdivisions of different sectors, analyzed individually, then analyzed across cases but within the same sector, and finally analyzed

across sectors. Both Confucianism-related values and SPM-related items are identified for code and theme generation. Thematic content analysis relies on both manifest and latent levels of analysis. At the latent level, the researchers' expertise in the field of Chinese management also helps with the data's interpretation [88]. The reported case data, both the case reports and the interviews, are read and re-read multiple times and iteratively for the code and theme generation. Narration and interpretation are used rather than software for frequency counting.

## 4. Results

This section presents the results derived after examining the 20 cases individually, across cases, and across industries. We identify three categories of sustainable SPM that emerge from Confucianism within Chinese enterprises: responsible people management, paradoxical people management, and humanistic people management (see Table 2). The following subsections discuss these results in detail.

### 4.1. Responsible People Management

*The Book of Mencius* provides an interesting value foundation that forms the basis for constructing Chinese managerial theory, including responsible management intended to achieve societal well-being and ensure harmony and high moral value within society, starting with leaders' consciousness [89]. These are interrelated elements that correspond to responsible management intended to ensure society's well-being through ethical consideration informed by the Confucian perspective on social harmony and morality, integrating corporate social responsibility as part of the ethical culture and reflecting the importance of caring for people, not only in terms of the firm's employees (HRM), but also others within the ecosystem (e.g., community, clients, suppliers). Indeed, responsible people management is concerned with being aware of societal well-being as an aspect of a firm's responsibility and then promoting it in reality.

Leadership's consciousness of societal well-being as an aspect of firm responsibility is the driver that triggers this sense of responsibility escalating at different organizational levels to render social sustainability part of sustainable management within the firm. In one of our studied cases, Alibaba, in addition to the illustrative quote concerning his awareness of the responsibility of social development, founder Jack Ma placed ethical value at the core of the management of the fast-growing digital business. During several critical expansion moments when large investments in the e-commerce industry in the rapidly growing market in China also meant opportunities for corrupt behavior to occur, both internally with managers and externally with suppliers and clients, while many viewed such corruption as normal in emerging markets such as China and tended to accept it with silence and discretion, Ma sacrificed high-performing managers involved in scandals and their high-level executives for not being able to monitor and manage their behavior. In the industrial context of real estate and construction, Tecsun's, Vantone's, and Vanke's top managers also clearly stated the relevance of morality in society and in their organizations due to corrupt behaviors being quite common in emerging markets. Neusoft's leader was more concerned with the larger responsibility for the well-being of more people as the firm grew than perceiving more power. In the notes of the researchers from their interview with the founder of Li-Ning, moral and ethical leadership were also remarked on.

**Table 2.** Sustainable strategic people management via Confucianism.

| Type of Sustainable SPM | Definition | Confucian Features | Illustrative Examples |
|---|---|---|---|
| Responsible management | To be aware of societal well-being as an aspect of a firm's responsibility and promote it | - Leaders' consciousness<br>- Social harmony<br>- Social morality | Ex. 1: "Today, big companies must solve social problems if they are to solve their own problems. Only then can the company last forever. Alibaba has tried to solve social problems such as unemployment and to answer needs for innovation. It is not simply a company, but also an ecological system". —Founder, Alibaba<br>Ex. 2: "The first and most important responsibility is to give (customers) the best products. Without that, other responsibilities are basically empty talk. The second responsibility is to treat employees well. Employees have chosen to work for a company in the hope that they can be happy and grow there… After fulfilling the first two responsibilities, if entrepreneurs still have financial resources left, they can invest in public undertakings". —Founder, Tecsun<br>Ex. 3: "From the interview, we sense an extraordinary humble man who leads by personal example, with passion, discipline, perseverance, moral character, and self-reflection. Though humble, he is also a man with great inner strength and a high level of moral integrity". —Researcher, Li-Ning |
| Paradoxical management | To innovate in times of uncertainty and ambiguity to combine two apparently distinctive aspects together | - Great learning (learning, judging, and creating)<br>- Openness<br>- Innovative organization | Ex. 1: "For Dongsheng Chen, to innovate is to imitate first. This is… the knowledge conversion process from others, like explicit and tacit knowledge, is spiral and transcendental … As part of innovative imitation, through learning first from diverse experiences in the West and other Great China regions, Taikang's own models have been built through the fusion of different pieces of knowledge to create something new…"—Commentator, Taikang Life Insurance<br>Ex. 2: "We Chinese like to use some Western ideas while still basing our operations on Chinese theories, critically absorbing the good part (of both the East and the West)". —Founder, Vanke<br>Ex. 3: "The innovative business model and strategic management of Fosun also provide insights for the potential integration of… the theory of Chinese management… Rooted in Chinese culture, traditions, and markets, Fosun integrates Western management conceptualizations holistically as an inseparable aspect of global leverage". —Commentator, Fosun Group |
| Humanistic management | To ensure the harmony of humans and nature for the common good | - Benevolence in people management<br>- Self-realization<br>- Positive attitude | Ex. 1: "Alibaba will eventually become the best money-making company, but I worry about losing our human sense. I want my company to be like a person, with feelings, consciousness, and a code of conduct". —Founder, Alibaba<br>Ex. 2: "Coming from a small town in Sichuan province and sharing equal awareness and commitment of 'good virtue', the founder, Zhang Yong, has made it clear that 'Changing fate with one's own hands' is the core value. What he wants to convey is that employees can achieve success through their hard work, dedication, and integrity… Haidilao is a platform through which to help employees achieve their dreams". —Researcher, Haidilao<br>Ex. 3: "Resources can be exhausted; only culture can be renewed. All industrial products are created by the intelligence of human beings. Huawei has no natural resources to rely on for survival; hence, we could only cultivate, in the human mind…"—Founder and CEO, Huawei |

The consciousness of leadership seems to be an essential element of responsible management, which is coherent with Confucian thought, even though it is not necessary for only high-level managers. Indeed, wise leaders can be distributed throughout an organization. In the cases of both Haidilao and Kidswant [5,7], the data clearly addresses how frontline employees are empowered to take the initiative and authorized to act as leaders, being responsible for their own actions. In Kidswant, shop staff are responsible for offering the best product recommendations to customers in the sense of the first responsibility that Tecsun suggests. In Haidilao, restaurant staff have the right to give discounts or gifts directly to customers without the need for authorization by their supervisors, with the aim being for them to be responsible for providing the best experiences and services to customers. Beyond this business level of responsibility, Haidilao staff often develop a community relationship with customers and participate in it in their private lives, such as by helping customers with matters outside the business context, thereby promoting social harmony and morality.

When responsible management philosophy is borne in mind by leaders, the implementation of the associated ethical values within organizations is intended to generate social harmony through the participation of all organization members. Otherwise, it is most likely more on-paper talk than reality. For instance, the founder of Tecsun places responsibility on general society after responsibility to customers and employees. It is in the third order and only manifests when entrepreneurs have the extra financial resources required to carry out socially responsible actions. Responsible management may encounter many types of challenges due to the economic costs involved in performing these ethical actions, which not only concern the inputs for corporate social reasonability programs but also the hidden costs and opportunity costs of losing projects or clients when implementing ethical values such as forbidding bribery or firing a high-performing salesman who engages in incorrect behavior. As indicated by the data, corruption is a common phenomenon in emerging markets such as China. Engaging in socially ethical actions in the real business context may prejudice economic performance, which may eventually result in economic sustainability risks. When the social dimension of sustainability conflicted with economic sustainability, several top leaders from the studied Chinese enterprises clearly stated that if their company needed to survive economically by sacrificing social ethical values, then it was preferable to go bankrupt than to survive unethically. The Confucian idea of righteousness lies at the center of these leaders' attempts to transfer this teaching to the rest of their organizations and ensure congruence between their personal and organizational values and objectives.

*4.2. Paradoxical People Management*

When there is a potential conflict between the economic dimension of sustainability and the other two dimensions (i.e., social and ecological), firms tend to sacrifice the social and ecological dimensions in favor of economic performance, or at best, to try and balance the three dimensions in pursuit of what is referred to as the triple bottom line [20]. Not only does the triple sustainability strategy need to be managed with the consciousness of leaders and escalated to different organizational levels, but it is also associated with a type of paradox that exists in many dimensions of management and requires both effort and attention. Paradoxical people management involves innovating in times of uncertainty and ambiguity to combine two apparently distinctive aspects. This is a process of great learning (i.e., to learn, judge, and create). Openness is essential for it to occur and create an innovative organization rather than being limited to an established frame and uneasy about dealing with two apparently confronting elements.

Many of the studied Chinese firms exhibit paradoxical management within a dual system due to their ability to deal with two opposing forces or contradictory requirements simultaneously. In addition to the quotes presented in Table 2, the top manager at Taikang also stated that "China is a big country with huge regional, cultural, and climatic differences, so we must allow a balance between consistency and flexibility. We need to pass

on our company's core cultural values, but to compete in the market, we must also respect special features of different markets". Commentators from Taikang suggested that such innovative learning is relevant and characteristic of emerging markets such as China, where uncertainty and ambiguity are characteristic and there is no clear rule. Similarly, Fosun's leader also stressed the need to "discover the co-existence of [the] interests" underlying conflicts and rivalries rather than eliminating them. By paradoxically managing philosophical values so as to operationalize them through leaders and learning processes, organizations can be open to learning, reflective with regard to judging, and creative in terms of innovating in a dynamically uncertain and ambiguous environment such as China.

On the one hand, the leaders from firms such as China Merchant Bank, Haidilao, and Huawei reflected on their learning processes, either from Western or Chinese wisdom or from business peers, when it comes to making judgments and creating new forms of business over time. Often, during this reflective creative process, paradoxical elements coexist and ways are found to merge them, thereby creating an innovative business model. In an apparently distinct way of doing business, successful Chinese enterprises have proven good at learning the apparently opposing management styles adopted in the West and merging them to create an innovative approach. In the words of the founder of Vanke, Chinese businesses "like to use some Western ideas while still basing our operations on Chinese theories, critically absorbing the good part (of both the East and the West)". A similar observation was made by a commentator from Fosun, namely that the innovative business model of Fosun is rooted in Chinese culture, traditions, and markets, although it integrates Western management and holistically conceptualizes it as an inseparable part of its business. Another example can be seen in the case of Huawei, where its founder brought in a Western consulting team during the growth phase to reform the supply chain and claimed to shape the (Chinese) feet to fit the (Western) boots. Lenovo also learned over time to create its own housing model for people management. Moreover, according to its founder, the process of acquiring IBM's personal computer unit was a process of learning, comprehending, and then generating a unique approach to global management to achieve balanced management of international talent and global business.

On the other hand, an innovative organization is built by innovative individuals who deal with paradoxical management in their daily business lives. This individual level of learning and creation is extended to the organizational level to foster innovativeness and sustain economic performance. Additionally, it is extended through social and ecological sustainability, as was reported in many cases. Haidilao's success as a restaurant chain was mainly attributed to its innovative business model and offer of extraordinary services such as free beverages and entrance and the availability of a kids' playground while waiting for a table. This paradox manifests in the cost of free services versus excellent services, in employees' power versus control over employees' inappropriate behavior, and in low-profile employees versus humanistic people management. Behind this innovativeness on the part of Haidilao was the coexistence and management of values and random management control. The complementary system encouraged employees to unleash their creativity and initiative to ensure better functioning at all levels and all types of tasks in an economically incentivized organization, and the core values of hard work and transforming lives through self-realization flow through the organization and interact with surrounding communities and stakeholders. Haidilao's employees were also encouraged to constantly provide creative ideas for organizational improvements and make friends with customers. Thus, an open and innovative organization is created through a great learning process and the innovative management of all these paradoxes.

### 4.3. Humanistic People Management

An individual's love of learning drives their need for constant self-cultivation and self-realization, accompanied by a positive attitude. *The Analects of Confucius* state that "if a man in the morning hears the right way, he may die in the evening without regret" [90]. In essence, this represents a learning organization in the context of Chinese enterprises,

where sharing knowledge and working in teams are important to promoting a learning environment within the organization and navigating through the uncertainty and ambiguity featured in the emerging market context. In addition, certain ambiguity and flexibility help create an appropriate climate for constant learning within the firm. In general, at the societal level, as *The Great Learning* points out, it "is to illustrate illustrious virtue; to renovate the people; and to rest in the highest excellence" [91]. Having an open social environment and creating a learning community can foster learning based on personal effort and organizational support. Being benevolent with regard to people forms the foundation for such humanistic management, or what the common good paradigm refers to as sustainability [20]. The harmony of humans and nature in terms of management essentially involves ecological sustainability and consideration of governance for the people, while a people-orientation is adopted as a practical purpose of Confucian thought rooted in a nature-orientation [79]. Thus, humanistic people management is concerned with the management of the organization through ensuring the harmony of humans and nature for the common good, with benevolence in relation to people serving as a means to self-realize with a positive attitude.

Although clearly economic values were upheld to tactically motivate employees to behave in accordance with corporate values, social values and ecological values were also strongly supported as long-term strategic missions in the studied cases. For instance, the individual goal combination (IGC) model adopted by Haier reflects this humanistic and ecological view of management. While the individual here refers to an employee, the goal refers to user demand to develop a win-win situation in which employees create value through having the autonomy to make decisions based on changes in the market, while employees also have the right to determine their income in line with the value they create for users. The essence of the IGC model is that the transformation of the independent IGC entities empowers them to rotate around users to create an ecosystem characterized by flexibility and responsiveness to the market rather than a traditional top-down triangulation. For Hisense, the key principle is that the company belongs to its employees, while its growth or failure is also closely connected to them. Moreover, the concept of service in the context of Hisense is not only concerned with consumers and their communities but also with employees and international subsidiaries, creating perfection and serving society, as described in its code of conduct.

The illustrative quotes in Table 2 also demonstrate how Alibaba is more concerned with the human side of the organization than with making money. Indeed, an Alibaba HR manager stated that they did not determine key performance indicator (KPI) progress based on sales performance or similar, instead using a survey to measure the happiness of employees. He continued by suggesting that a happy employee will transmit their positive attitude to customers and the environment. In fact, in multiple interviews, Alibaba was said to place employees ahead of shareholders. In the case of Haidilao, a large number of employees came from the region where the founder of the company was born. Haidilao built schools in the area to facilitate the children of employees having a better educational environment, and the company also paid a monthly sum to the parents of employees to help improve their quality of life. This represents an example of adapting to the particularity of the Chinese context, where employees from rural areas often have to leave their family behind and move to the city to secure a good salary and improve their lives. In addition to being a socially responsible action that the company undertakes, this approach reflects the essential philosophical value of harmony and the common good for people outside the organization but within the ecosystem in which people self-realize with a positive attitude and flourish in a large family and communicative context. This latter character also reflects the righteousness and congruency between leaders' personal values and objectives and organizational ones. The founder of Huawei also emphasized the relevance of the human mind as a resource that can be exhausted, while culture can be renewed, and that "industrial products are created by the intelligence of human beings".

The example of the three sage kings from ancient China illustrates how a successful leader's personal qualities and virtues are required to care for the common good and ensure people's well-being rather than exploit their position [92], as some of the above illustrations also show. The presence of these virtues and benevolence in people management escalates from the individual to the team, the organization, and society, generating ethical, ecological, and even economic effects as a humanistic managerial approach. At the individual level, humanistic management can bring the returns of commitment and involvement from people within the organization, in addition to improving their work-life balance [62,93]. Tecsun's leader not only limits the profitability of products to a reasonable level rather than taking advantage of market exploitation but also quite strictly restricts the behavior of employees through good conditions and terms that might seem out of reach of organizations from a Western perspective. Kidswant empowers employees to interact with customers and builds a holistic business model to combine services and products. Similar to many of the other studied enterprises, including Alibaba and Haidilao, the company believes that employees need to be happy, and this happiness can be extended to customers and the community, eventually allowing the firm to be economically sustainable after achieving ecological and social sustainability.

### 4.4. Integrative Sustainable People Management via Confucian Values

As a result of our analysis of the 20 successful Chinese enterprises, we propose the following framework for sustainable people management via Confucianism (Figure 1). Taking the triaxial model of MBV theory as our theoretical foundation, we derive a three-dimensional value model via the functioning frame of SPM and Confucianism to achieve the three pillars of sustainability: social, economic, and ecological sustainability. MBV is a generic value-based (i.e., ethical) management model that has been theoretically proposed and empirically validated by Dolan et al., including in the Chinese context [9,62,94,95]. We extend the proposed theoretical framework to a sustainable SPM view that is inclusive and broad at the highest level of organizations. This MBV model concentrates on the three core axes of organizational cultural value: economic, ethical, and emotional [62].

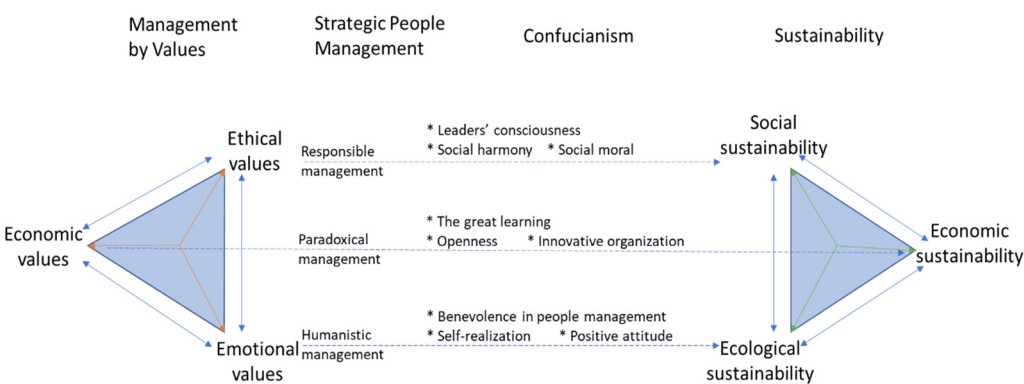

**Figure 1.** Sustainable strategic people management via Confucianism.

Three types of SPM were identified in relation to the sustainability development and management of firms: responsible management, paradoxical management, and humanistic management. These three types of sustainable SPM have been discussed in the preceding subsections, which included illustrative case examples. This simplified framework offers the explanatory power of the ethical value driving management toward sustainable growth via SPM combined with the Confucian perspective in the studied Chinese enterprises. Given the increasing interest in understanding sustainability in a global context, the sustainability literature is growing and moving in the direction of maturity. The proposed framework connects the value foundation of management with the three dimensions of sustainability: economic, social, and environmental/ecological sustainability [20]. In particular, it adds value to the inner management of an organization from a sustainable SPM

perspective based on Confucian values, serving as a reference point for companies from other emerging markets seeking to engage in further exploration and exploitation.

## 5. Discussion, Conclusions, and Limitations

### 5.1. Conclusions and Propositions

As this study explores the interconnection between sustainable SPM and Confucianism in the cases of Chinese enterprises, the findings reveal three types of sustainable SPM: responsible SPM, paradoxical SPM, and humanistic SPM. Conceptually, this study also establishes the interlinkage between the value system and SPM in contributing to the sustainable growth of firms in multiple dimensions via Confucianism. The 20 studied cases also illustrate the potential and feasibility of the common good paradigm beyond the triple bottom line aspect [20]. This is especially relevant to the current debate concerning the relation between sustainability and people management. Even though the triple bottom line model recognizes the intertwined relations among the three dimensions of sustainability [96], it fundamentally adopts the concept of traditional economic gain as the principal business purpose while accommodating external pressure on the social and environmental dimensions [20]. By contrast, the common good approach perceives the sustainability challenges as the fundamental responsibility of the firm, as it needs to self-evolve and grow sustainably in the long term in an eco-system context rather than independently [20,97]. Our analysis indicates that even though some successful Chinese enterprises adopt a triple bottom line perspective (e.g., Tecsun), others apply the common good perspective (e.g., Alibaba). These findings add value to the literature concerning sustainability in terms of management and organizations, with an emphasis on people-related management in a highly dynamic VUCA context.

The incorporation of Confucian values into the sustainable SPM framework transforms values into strategic actions associated with the three dimensions of sustainable development: ethical sustainability, economic sustainability, and ecological sustainability. The identification of these three types of sustainable SPM could be relevant for future people management studies addressing the development of the common good paradigm in relation to people. The interconnection between sustainable SPM, Chinese management, and Confucianism also showcases the theoretical generalization of Eastern cultural values to the sustainability paradigm. Further Eastern contexts and additional value studies may be desirable in relation to sustainability research concerning management and organizations, with a special focus on people. Traditionally, economic performance has been the principal outcome measure of the firm, including in the field of SHRM. Other measures in this regard could include financial measures [98] or productivity and quality measures [99,100]. Given the disagreement regarding the meaning of performance in the organization literature [101], a solely economic-oriented measure of firm performance could misrepresent and mislead a firm's behaviors, including those of organizational members. Due to the triple bottom line for people management and especially the common good paradigm [20], incorporating the three dimensions of sustainability is timely and important. Our study incorporating Confucian values into successful Chinese management provides empirical evidence of this approach's feasibility and effectiveness in highly dynamic VUCA contexts. Most of the studied Chinese enterprises place emphasis on social and ecological sustainability equal to or ahead of economic priority.

This case analysis, based on Confucian values and SPM, is grounded in MBV theory. Three types of sustainable SPM can be identified from the studied cases: responsible people management, paradoxical people management, and humanistic people management. Our research demonstrates the effectiveness of a firm orientation centered on people when it comes to generating social sustainability, ecological sustainability, and consequently, economic sustainability, rather than viewing the ecological and social dimensions of sustainability as costs that require a trade-off with economic sustainability. This understanding highlights the need for certain revisions to management assumptions, which are partly reflected in the following propositions for future studies:

**Proposition 1.** *Sustainable SPM situates human beings as the center of attention in relation to firm management. Humanistic people management is not concerned with treating people as a resource, an asset, or a capital that can be calculatedly managed from an economic perspective; rather, it is an ecological perspective on harmoniously managing the relations between humans and nature, that is, their surrounding environment;*

**Proposition 2.** *Sustainable SPM considers ethics to be the foundation of firm management. Responsible people management must start with leaders' consciousness and situate righteousness at the core of corporate values to guide all stakeholders' behaviors and actions rather than assigning economic performance priority, even if doing so comes at the cost of other sustainability dimensions;*

**Proposition 3.** *Paradoxes exist in the ecological system of human beings, as in the case of firm management. Paradoxical people management deploys people as knowledge workers to learn, judge, and create innovative organizations, as well as to develop new ways of doing things in ambiguous and uncertain environments such as emerging markets, to sustain the co-development of several apparently contradictory elements.*

*5.2. Discussion and Limitations*

Although general sustainability is considered within the three dimensions of economic, social, and environmental/ecological sustainability, prior studies have extended it to six dimensions by adding managerial, governance, and infrastructure sustainability in certain fields, such as sustainable tourism [102]. Other studies have focused on sustainability performance and its impact with regard to different stakeholders [103]. This study focuses on sustainable SPM as a form of inner management for firms via Confucian values. Mencius asked King Hui of Liang why His Majesty used the word "profit", and he continued by criticizing the potential hurt it could bring [89]. Coincidentally, in relation to modern management, Grant has raised the dilemma between the shareholder approach and long-term firm profits using the case of Boeing [104]. Here, solely focusing on economic values can be harmful to an organization's long-term profits, while equilibrium with ethical and emotional values is necessary for the strategic development of both the organization and its people. In this study, Liu Chuanzhi, the founder of Lenovo, also observed that American top managers in IBM's post-acquisition personal computer unit had been focusing on short-term profit generation to ensure their economic performance, which eventually hurt the long-term sustainability of the firm due to it not adequately investing in strategic needs.

The consciousness of leaders in terms of the non-economic dimension of sustainability or performance represents the starting point for being aware, learning, judging, and innovating, while responsible people management forms the foundation necessary to achieve social sustainability. People management can contribute to social responsibility and environmental sustainability [105]; that is, responsible management and ecological management contribute to sustainable development in combination with humanistic management. In a highly dynamic VUCA context, firms need to function as a self-organizing complex system to confront uncertainty and ambiguity, while knowledge workers serve as a humanizing strategy in such a VUCA world [106]. This human-centric view of the firm accords with the perspective of Kalton, who considers that to differentiate from others, humans hold experiential intelligence and progressively develop their consciousness to a complex level, which represents an evolving strategy for adapting to the environment [78]. While the organization and environment evolve and co-evolve, consciousness is key if the adaptive strategy is to fit the evolutionary ecosystem and help achieve ecological sustainability.

The rapid changes seen in China since its market opening have caught the interest of scholars, practitioners, and politicians seeking to better understand the Chinese management phenomenon [1,31]. A variation in the traditional economic firm performance judgment based on multiple sustainability dimensions seems to be the norm in successful Chinese enterprises, in addition to being compatible with Confucian values. The sustain-

ability focus that SPM strives for instead of a performance orientation (i.e., often referring to economic and financial outcomes) also characterizes the harmonious society that China and Chinese enterprises endeavor to foster [7,9]. This sustainable SPM perspective may be useful in China's dynamic environment and extendable to other emerging markets where the paradoxes between economic development and social and ecological development require urgent management. In the research context, this may also help with the refinement of performance measurement by encouraging more variations in sustainability measures and related performance measures, such as increasing the diversity of performance assessments as suggested by scholars in especially complicated and difficult international contexts [101,102,107]. Rather than financial measures such as the return on assets (ROA) and return on equity (ROE), there should be a shift toward sustainability measures, including "ecological, social and ethical considerations" [108], particularly under the paradigm of sustainable SPM development.

There are already many different economic and alternative measures available. The objective and subjective measures include HR performance and work-life value congruency [24,62,93,109], which can be used as references for measuring emotional or ecological performance. In terms of ethical performance, some indexes of corporate social responsibility and corporate governance, such as the Global Compact [110], could serve as references. Unlike other studies, this study does not attempt to materialize these specific measures for all three dimensions of sustainability [102]; rather, it aims to provide evidence of the reality of successful Chinese enterprises through applying Confucian values combined with the social and ecological sustainability dimensions to gain economic sustainability. In return for care from organizations through leaders, individuals exhibit a positive attitude and are increasingly committed to and involved in the organizational activities necessary to achieve organizational objectives.

This can be fundamentally differentiated from the mechanic view of the organization founded on the economic paradigm of management by considering that a firm, as an organic structure, facilitates stakeholders' learning inside and outside of organizations in an attempt to dynamically judge and create new forms of being and doing in order to better face global challenges [111,112]. *The Analects of Confucius* start with the following: "Is it not pleasant to learn with a constant perseverance and application?" [90]. In terms of the Confucian dynamics of learning, openness creates an innovative organization with great learning at the core of a dynamic and complex organic system, which allows it to evolve knowledge in a spiral manner inherited from human beings. The integration of the three types of sustainable SPM provides and sustains these dynamics inside the organization and its ecosystem. Further exploration of the fuzzy boundaries of such dynamism is required. As the conceptualization of SPM is relatively novel despite the people management concept being old, the number of empirical studies conducted from a quantitative perspective is currently limited. With the maturation of both the field and the conceptualization, further conceptual and empirical studies are necessary in the Chinese context as well as in other economic contexts, both emerging and advanced economies, to observe its effects on sustainable development.

Given the nature of the qualitative case study method, we do not present any statistical generalization of our empirical results. In fact, we expect a possible theoretical generalization of the proposed sustainable SPM model to be further tested via either qualitative or quantitative methods. That is, the results of this study are limited to theoretical generalization, not statistical generalization. Another limitation of this study is that we explore the interlinkage of SPM with sustainability through Confucianism in the Chinese enterprise context. As a consequence, we cannot fully perform a cross-divisional comparison between enterprises that apply the good common paradigm and those that apply the triple bottom line, which was discovered during the analysis of the present results. Future studies should purposefully select samples or cases with these two different profiles and then compare their determinants, processes, and consequences to expand our understanding of the theoretical development of sustainability. The quantitative research method is also

called for in future research in this direction to help improve its statistical generalizability. For instance, a cross-country comparison between sustainable SPM and the results of innovation (e.g., patents, trademarks, and designs) could explore and exploit the effectiveness of the sustainable SPM model in terms of innovative outcomes at the national level.

**Funding:** This research was funded by an IUJ Research Institute (IRI) Research Grant (Grant number IUJ2023-03).

**Institutional Review Board Statement:** Not applicable.

**Informed Consent Statement:** Not applicable.

**Data Availability Statement:** Data are available in the published books and cases stated in Section 3.2—Data Collection and Analysis.

**Acknowledgments:** Yuvita Andriana has provided technical support for some data collection, organization, and reference formatting.

**Conflicts of Interest:** The author declares no conflict of interest.

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
