# Peer review of "Sustainable Strategic People Management: A Confucian Perspective on Chinese Management"

_sustainability, doi:10.3390/su15129188_

Round 1
Reviewer 1 Report
Needs English editing. From the first lines "The paper explores...." The article cannot explore itself.
Incorrect sentence formation: "Based on twenty cases of successful Chinese enterprises, we further analyze these elements across five sectors." It is not clear how the cases are related to the elements.
The abstract needs significant revision. This is not an introduction. It should be stated what was analyzed, researched, discovered, proposed, etc.
The introduction is not written logically. It just looks like a set of sentences with quotes.
The expression "top economic power position" is very incorrect. Authors should then make a reference according to which classification the given gradation belongs to.
No logical connection between the sentences
"...While there are still continuous social political and economic issues that China needs to resolve to its sustainable development, one can witness its development in decades as factual data for its sustainability, largely attributed to its private business sector besides the institutional factors ( Nee & Oppers, 2012; Tsui, Zhang, Chen, 2017). Among different .."
2 "This is especially reflected during the COVID-19 period that in spite of a prediction of economic slowdown, the factual and projected data of China are still much more optimistic than those of the advanced economies in the overview of World Economic Outlook (International Monetary Fund , 2021; 2022).
While it is fascinating.."
This sentence "As a global concern, multinational corporations (MNCs), either large sized or small-medium-sized enterprises, also make their efforts to play a role and act toward sustainability and SDGs, despite the critique on the insufficiency..." is an unfortunate abridgement of a sentence from an article (R Van Tulder · 2021// https://www.ncbi.nlm.nih.gov/pmc/articles/PMC7884867/)
The authors do not understand "stylish development", "sustainable growth", "economic dimension", they identify them with the concepts of "profitability" and "efficiency" of entrepreneurial activity; "...To reflect the sustainability-focus of this study..."
Unreliable justification: "China took over the United States and became the largest economy in the world in 2014 based on the International Monetary Fund (IMF) (Bird, 2014)..." The authors refer not to the IMF, but to Bird, 2014. 2014 IMF statistics could not be published in 2014. Only in 2015. Further "This was a result of .... (BBC News China, 2012)). It seems that the BBC in 2012 substantiated that in 2014...
Some examples are given for an unclear reason: "Human resource management (HRM) is a well-studied field in the English-dominated management research and teaching, ..."
The whole text is just a set of quotes. The authors did not try to draw their own conclusions, everything boils down to "several findings are identified:.."
Methodology
"...1) being successful..." by what criteria is it determined, "...being successful in order to observe and interpret their strategic people management..." there are already publications that prove that there is a direct correlation between success and personnel management does not exist.
Data Collection and Analysis - absent.
Unfortunately, the text lacks any logic of the presented material, methodology, and own research as well.
Conclusions and the text itself are not correlated.
Reviewer 2 Report
A good research problem and a good attempt. Please see the attached file for my detailed comments

Reviewer 3 Report
The presentation reflects the present state of knowledge. The paper is very well structured. The Introduction section is good, in this section the authors present clearly the objectives and the main contributions of the study. The authors provided sufficient background and include relevant references. The results are clearly presented. The conclusions are supported by the results. Author should follow the format of the references in text.
Author Response
Thank you very much for your positive feedback. We are grateful for that. The reason that we didn’t do the reference format in text in the last version is that we expected that reviewers may request us to make significant changes. As the reference format requires significant manual work when there is change of reference position, we think it will be better to adjust the format in a late stage. Anyway, we now have modified the reference format of the journal in the revised version. Thanks a lot again for your kind words.
Reviewer 4 Report
The topic of the paper is very interesting, but is only a theoretical study, and only reminding some models, the paper in this state, in my opinion, is not recommendable to be published. The journal itself wants some results obtained by implementing an empirical study.
For example, in order to agree with the article publication, it is recommendable to link the model you developed, data on countries regarding the patents, marks, and designs (PMD), the VRIO architecture, that you reminded, the model linked by Sustainable Strategic People Management via Confucianism, and develop a new model, your own, based on forecasting and using simulation and mathematics, and observe in the next 5-10 years, future data for PMD model for China compared with other countries that you analyzed. Thus, you may then develop a strategy on long term use not only for your analyzed country, but for any entity who wants performance on long term by people and for people (as you are talking about people management).
I hope, this opinion will give you inspiration and the opportunity to find a way to continue your theoretical study with an empirical one.
The second piece of advice is to improve your references, as they are old and very old- it is indicated to add along with your 4 sources from 2022, others from the same year, at least 10-15 new and updated sources.
At this moment, the article needs major improvements, because it has serious flaws, additional experiments are needed, and research is not conducted correctly.
Author Response
Dear reviewer,
Thank you very much for your comments, feedback and time dedicated to review our manuscript for improvements. We are grateful to know that you think the study topic is interesting. We would like to clarify that this is a qualitative case method empirical paper, not a theoretical paper. We think it is our responsibility to improve the writing and expression if the earlier version made you perceive that. Now we have made a lot of significant changes in the abstract, introduction, methodology, results, and conclusion sections. We hope with all these improvements, it helps in clarifying your concerns.
As the research purpose is to explore the interlinkage of strategic people management with sustainability through Confucianism in the Chinese enterprise’s context, we could not accept your proposed method for our current study. However, we indeed reckon that is an interesting idea for future research. Therefore, we have worked more in the discussions and limitations section, and added your idea in this part for future research direction.
These ideas that you suggested are inspirational and we would like to continue in the future these lines of work.
Based on your suggestions, we have also added more than 30 new references. Most of them are of 2020 and forward, and some of them are of 2010s.
We have also significantly revised the text in terms of linguistic issue with the help of peers. We will also contract a professional service to edit English to avoid linguistic problem at the late stage with the final version of the manuscript. Thanks a lot!
Round 2
Reviewer 1 Report
The authors significantly revised the article. Edited the style of the article, and clarified speech remarks. However, authors cannot argue for the definition of certain terms based on the publication of one author. It is better to use generally well-known terms, which are used in the publications of international organizations such as the IMF.
In each national economy, the same term sounds and writes differently, it is not fashionable to simply translate it using universally recognized terms.
This is exactly what the comment about determining efficiency was all about. The authors' arguments are well-founded, but the level (national, regional or global) should be taken into account. Each level has its own system of indicators and their definition.
Author Response
Dear reviewer,
Thank you for your positive comments on the revised version. Following your suggestions, I have now revised all terms consistently to sustainable growth and economic dimension through the text.
Thanks again for your attention and time dedicated to the revision.
Reviewer 2 Report
I have no further comments. The authors' response to the queries was detailed and satisfactory.
Author Response
Dear reviewer,
Thanks for your positive feedback on the revised version of manuscript.
Best regards,
Reviewer 4 Report
I will maintain my opinion in order to improve the content of the article; is not so hard to improve it as the request:
„For example, in order to agree with the article publication, it is recommendable to link the model you developed, data on countries regarding the patents, marks, and designs (PMD), the VRIO architecture, that you reminded, the model linked by Sustainable Strategic People Management via Confucianism, and develop a new model, your own, based on forecasting and using simulation and mathematics, and observe in the next 5-10 years, future data for PMD model for China compared with other countries that you analysed. Thus, you may then develop a strategy on long-term use not only for your analysed country, but for any entity who wants performance on long term by people and for people (as you are talking about people management)”.
I hope the authors perceive the importance of developing their own model based on the data presented. Thus, the article is still with significant improvements.
Author Response
Dear reviewer,
Thanks for your comments. We have taken it very seriously and thought of how to proceed so. However, we did find it is very hard to improve it as your request.
You suggested to link the model developed with data on countries regarding the patents, marks, and designs (PMD), the VRIO architecture, with the model of Sustainable Strategic People Management via Confucianism, and develop a new model, based on forecasting and using simulation and mathematics, and observe in the next 5-10 years, future data for PMD model for China compared with other countries that you analysed.
The reason of this difficulty is that is a totally different research from what this current research addresses. We focus on the Chinese enterprises context to study qualitatively twenty cases to build a model based on thematic content analysis. It is a quite different research focus from what you suggested.
We realized that it may be because of the contextualization part of the earlier version of mansucript to highlight the successful performance of China in terms of sustsinable development through innovation, by using WIPO's cross-country data of patents, marks, and designs, misled you to this interpretation.
In order to avoid this further confusion, we now decided to delete this misleading Table of comparative data. As this is just a contextual illustration, we think it could be sufficient to make clarification in the text in a simpler way. In this manner, we hope it clarifies better and avoids misunderstanding.
We would like to emphasize that it is an empirical paper with qualitative case method. It is difficult to add another analysis of quantitative empirical study as the focus and research questions are totally different.
Thanks in advance for your attention.
Best regards,
Round 3
Reviewer 4 Report
The authors improved the study according to the requirements. It is also necessary to add some sources for paragraphs from the lines: 271-280 and from 343-354. Minor improvements are requested in order the article to be published.
Author Response
Dear Reviewer,
Thanks a lot for your detailed comments and suggestions on the points of changes for improvements. Now I have done so in the suggestion paragraphs:
(1) Add some sources for the lines: 271-280 (in the revised version it is lines 310-319).
(2) Add some sources for lines 343-354 (in the revised version it is lines 389-402).
Best regards,